# Aristoteline, an Indole-Alkaloid, Induces Relaxation by Activating Potassium Channels and Blocking Calcium Channels in Isolated Rat Aorta

**DOI:** 10.3390/molecules24152748

**Published:** 2019-07-29

**Authors:** Fernando Romero, Javier Palacios, Ignacio Jofré, Cristian Paz, Chukwuemeka R. Nwokocha, Adrián Paredes, Fredi Cifuentes

**Affiliations:** 1Vicerrectoría de Investigación y Postgrado, Programa de Doctorado en Ciencias Médicas, Universidad de la Frontera, Temuco 4780000, Chile; 2Laboratorio de Bioquímica Aplicada, Departamento de Química y Farmacia, Facultad Ciencias de la Salud, Universidad Arturo Prat, Iquique 1110939, Chile; 3Laboratorio de Neurociencias y Biología de Péptidos, Centro de Excelencia en Biotecnología de La Reproducción, Facultad de Medicina, Universidad de La Frontera, Temuco 4811230, Chile; 4Laboratorio de Productos Naturales y Descubrimiento de Fármacos, Departamento de Ciencias básicas, Universidad de La Frontera, Temuco 4811230, Chile; 5Department of Basic Medical Sciences Physiology Section, Faculty of Medical Sciences, The University of the West Indies, Mona, Kingston 7, KGN, Jamaica (W.I.); 6Laboratorio de Química Biológica, Instituto Antofagasta, Universidad de Antofagasta, Antofagasta 1270300, Chile; 7Laboratorio de Fisiología Experimental, Instituto Antofagasta, Universidad de Antofagasta, Antofagasta 1270300, Chile

**Keywords:** Aristoteline, vascular activity, calcium channels, potassium channels, rat

## Abstract

Alkaloids derived from plants have shown great medicinal benefits, and are often reported for their use in cardiovascular disease management. *Aristotelia chilensis* (Molina) Stuntz (Maqui) has shown important medicinal properties in traditional useage. In this study, we evaluated the effect of the indole-alkaloid aristoteline (ARI), isolated from leaves of Maqui, on vascular reactivity of isolated aortic rings from normotensive rats. ARI induced relaxation (100%) in a concentration-dependent manner in intact or denuded-endothelium aortic rings pre-contracted with phenylephrine (PE; 1 μM). However, a specific soluble guanylyl cyclase inhibitor (ODQ; 1 μM) significantly reduced the relaxation to ARI in aortic rings pre-contracted with PE. In the presence of ARI, the contraction induced by KCl or PE was significantly (p < 0.05) decreased. Interestingly, the potassium channel blockade with 10 μM BaCl_2_ (Kir), 10 μM glibenclamide (K_ATP_), 1 mM tetraethylammonium (TEA; KCa1.1), or 1 mM 4-aminopyridine (4-AP; Kv) significantly (*p* < 0.05) reduced the ARI-induced relaxation. ARI significantly (*p* < 0.05) reduced the contractile response to agonist of Ca_V_1.2 channels (Bay K8644; 10 nM), likely reducing the influx of extracellular calcium through plasma membrane. The mechanisms associated with this process suggest an activation of the potassium channels, a calcium-induced antagonism and endothelium independent vasodilation that possibly involves the nitric oxide-independent soluble guanylate cyclase pathway.

## 1. Introduction

Aristoteline (ARI) is an indole-alkaloid derived from *Aristotelia chilensis* (Molina) Stuntz, a tree of the Elaeocarpaceae family that is endemic to Chile and popularly known as the Maqui berry [1]. The fruits and leaves of *A. chilensis* have shown an important range of biological properties including antioxidant [2], anti-inflammatory [3] and analgesic effects [4]. Leaf extracts have been used in alternative and folkloric medicine due to its analgesic and anti-inflammatory properties that are used to alleviate oropharyngeal discomfort [4]. There are a few studies about the vascular response of pure compounds from *A. chilensis*.

The chemical analysis of the *A. chilensis* shows that it has constituents such as flavonoids in fruits [5,6], and non-iridoid monoterpene indole-alkaloids and polyphenolic compounds in leaves [7,8]. *A. chilensis* leaves have been reported as a source of several indole-alkaloid rich compounds, including aristoteline/aristotelinine [9], aristotelona [10] and aristona makonina [11]. The methanol extract from fruits of *A. chilensis* reduced oxidative damage caused by the release of free radicals after ischemia reperfusion [6], and 8-Oxo-9-dihydromakomakine a tetracyclic indole-alkaloid extracted from leaves of *A. chilensis* reduced the contractile response to agonists of Cav1.2 channels in rat aorta [12]. It has been hypothesized that these biological effects could be useful for the treatment of vascular diseases such as hypertension [13].

It is possible that these pure compounds may modulate the vascular response because alkaloids like rutaecarpine, naucline, curine, glypetelotine cause vasorelaxation of vascular tissues through the blocking of Ca_v_1.2 channels in vascular smooth muscle [14,15]. Indole-alkaloids derived from *A. chilensis* will warrant investigations to ascertain its role in the management of cardiovascular ailments. Thus, the study aims to evaluate the effects of ARI on vascular reactivity and elucidate the mechanisms associated with the ARI-induced relaxation/contraction in the isolated aortic rings of rats.

## 2. Results

### 2.1. Chemical Composition

Isolated crystals from the leaves of *A. chilensis* were chemically characterized as ARI by using 1D and 2D NMR. They showed a high degree of structural similarity [9] as they were composed of an indole moiety with a planar surface and attached to an N-aliphatic system, forming five rings (Figure 1). The chemical formula of the compound was C_20_H_26_N_2_ [16].

### 2.2. Relaxation Effect of Aristoteline (ARI) on Isolated Rat Aorta

The different concentrations of ARI were added in accordance with the cumulative concentration model, which induced a decrease in isometric tension in aortic rings pre-contracted with phenylephrine (PE) (1 μM). This relaxing effect was observed on rings with and without endothelium (Figure 2A). This result was confirmed because ARI also produced relaxation in aortic rings pre-incubated with 100 μM N^ω^-nitro-l-arginine methyl ester (l-NAME; a nitric oxide synthase inhibitor) (Figure 2B). However, the inhibition of soluble guanylyl cyclase with 1H-(1,2,4)oxadiazolo[4,3-a]quinoxalin-1-one (ODQ) significantly decreased (*p* < 0.05) the ARI-induced vasodilation in aortic rings versus control (Figure 2C). The IC_50_ to ARI significantly raised (*p* < 0.05) in the presence of ODQ (Table 1).

### 2.3. ARI Reduced the Contractile Response to KCl and Phenylephrine (PE)

The effect of ARI on the contractile response to KCl and PE of rat aortic to rings was explored. The pre-incubation with ARI (10 μM) significantly (*p* < 0.05) reduced the maximal contraction to 60 mM KCl (125 ± 3% control vs. 84 ± 2% with ARI; Figure 3 and Figure 4A). The EC_50_ to KCl significantly (*p* < 0.05) increased in the presence of ARI versus control (Table 2). The maximal response to 10 μM PE significantly (*p* < 0.05) decreased (166 ± 11% control vs. 114 ± 5% with 10 μM ARI; Figure 4B). The EC_50_ to PE significantly (*p* < 0.05) raised in the presence of ARI versus control (Table 2). Caffeine was used to analyse the effects of ARI with a methylxanthine alkaloid comparatively; 10 mM caffeine significantly (*p* < 0.05) reduced the maximal contraction to 60 mM KCl (125 ± 3% control vs. 83 ± 1% with caffeine; Figure 4A) and 10 μM PE (166 ± 11% control vs. 9 ± 1% with caffeine; Figure 4B). The EC_50_ to KCl and PE significantly (*p* < 0.05) increased in the presence of caffeine versus control (Table 2).

### 2.4. Role of Extracellular Calcium in the Vascular Response to ARI

Pretreated aortic rings with ARI (10 μM) were stimulated with PE (1 μM) in order to induce a contractile response in the calcium-free medium. In the first instance, the contractile effect generated by PE on the aortic rings pretreated with ARI was significantly lower (*p* < 0.05) in calcium-free medium than that observed in the control (Figure 5A). Then, the cumulative addition of extracellular calcium in the aortic rings of the control group showed a significant elevation in the tonic response, but not in the aortic rings pretreated with ARI (Figure 5B). The Figure 5B shows the contractile response after restoring extracellular calcium (0.1 mM to 1 mM). The contractile response to PE at concentrations of 1 mM CaCl_2_ was significantly lower (*p* < 0.05) versus control.

As shown in Figure 5C, we confirmed that 10 μM ARI significantly reduced (*p* < 0.05) the contractile response to 10 nM Bay K8644, an agonist of Ca_V_1.2 channels, versus control.

### 2.5. Role of Potassium Channels on Vascular Relaxation of ARI

To study the role of potassium channels on the vascular relaxation, aortic rings of rat were pre-incubated with BaCl_2_, glibenclamide, 4-aminopyridine (4-AP) and tetraethylammonium (TEA).

At ARI concentrations of 1 μM to 100 μM, there were significant differences with respect to the control curve, where relaxation was lower in the presence of the 10 μM BaCl_2_ (*p* < 0.05; Figure 6 and Figure 7A). The IC_50_ to ARI raised significantly (*p* < 0.05) in the presence of BaCl_2_, glibenclamide, 4-AP and TEA versus control (Table 1; Figure 7). The findings above of ARI were confirmed by the reduction of the vascular contraction induced in presence of KCl, BaCl_2_ or TEA (Figure 8 and Figure 9).

Although a reduction of the basal tone following washout of KCl was observed (Figure 8) which could be explained by the stretch-induced calcium during washout of the tissue [17], the basal tone decreased in a similar value in both aortic rings and they were brought to an identical basal tension before the continuation of the experiment. Normalization may not always correct for erros, and that was why we have converted all contractions and relaxations to a percentage. That way, we can truly present the degrees of contractions and or relaxations as a true representation of our experiments.

### 2.6. Role of Prostaglandins on Vascular Relaxation of ARI

To provide some information about the endothelial factor involved in the relaxation response to ARI, the vascular rings were pre-incubated with indomethacin (10 μM) for 20 min before the addition of PE. Indomethacin significantly enhanced the relaxation to ARI versus control. The IC_50_ for ARI significantly reduced (*p* < 0.05) in the presence of indomethacin (Table 1; Figure 10).

## 3. Discussion

The results obtained suggest for the first time the vasodilator proprieties of ARI, an isolated indole-alkaloid from *Aristotelia chilensis* (Molina) Stuntz (Maqui berry) leaves. Results showed that ARI-induced an independent relaxation of the endothelium, which involves the activation of the potassium channels, and the reduction of the calcium influx through Ca_V_1.2.

The relaxation activity of vascular endothelium and the endothelial nitric oxide synthase (eNOS) were studied. ARI caused relaxation in intact and endothelium-denuded aortic rings when they were exposed to cumulative concentrations of the extract. The inhibition of nitric oxide synthase with l-NAME did not reduce the relaxation mediated by ARI. However, we observed that the ARI-induced relaxation was blunted by pre-incubation with ODQ in aortic rings. Traditionally, nitrovasodilators are used in the activation of sGC, but a few alkaloids, such as ergotamine, dihydroergotamine and other compounds (YC-1, a benzylin-dazole derivative, or BA 41-2272) can stimulate nitric oxide-independent soluble guanylate cyclase activity and thus produce vasodilation [18,19,20].

The activation of potassium channels in the vascular smooth muscle cells causes membrane hyperpolarization, and vasodilation [21]. The pre-incubation with BaCl_2_ (no-selecive blocker of inward rectifying potassium channels; Kir), glibenclamide (K_ATP_, an ATP-sensitive potassium channel blocker), TEA (non-selective blocker KCa1.1 channels) and 4-aminopyridine (4-AP; non-selective voltage potassium blocker channel, Kv) significantly diminished the ARI-induced relaxation and the contractile response to KCl, BaCl_2_, and TEA versus control in aortic rings. 4-AP has greatest potency at Kv1 and Kv3 family members at micromolar concentrations, and also blocks Kv2 and Kv4 at millimolar concentrations [22]. As such, ARI-induced vasodilatation may be modulated by actions of the potassium channels (Kir, K_ATP,_ Kv, KCa1.1).

The results also showed that endothelial prostanoids could be involved in the ARI-induced relaxation. This would explain why indomethacin significantly enhanced the relaxation to ARI in aortic rings pre-contracted with PE. Other alkaloids, such as rotundifolone, have been reported to be partially dependent on prostaglandin production pathways, and endothelium dependence vasoreactivity [23]. Alternatively, there may be that the PE releases contractile endothelial factors like prostaglandins [24], and then, the ARI-induced relaxation is independent of prostaglandins.

In our calcium dependence studies, the contractile response to PE was reduced in the presence of ARI in a free calcium medium, and by extracellular calcium addition. This observation was confirmed by other experiments. First, the pre-incubation with ARI reduced the contractile response to the influx of calcium from extracellular sources through Ca_V_1.2 as showed in the result of the Bay K8644 in rat aorta. Second, the pre-incubation with ARI significantly reduced the contractile response to PE, and KCl versus control. Previous studies had demonstrated that dicentrine, nantenine (aporphine alkaloids) and tetrandrine (a bis-benzylisoquinoline) reduced the contractile response to KCl or α-adrenergic agonists [25,26,27].

Third, the pre-incubation with caffeine (a methylxanthine alkaloid) activates the ryanodine receptor [14], and then, the release of calcium from intracellular stores blunted the contractile response to PE. Vascular smooth muscle contraction induced by PE in the absence of extracellular calcium is due to the release of calcium from the intracellular stores through the synthesis of IP_3_. Restoring extracellular calcium causes an additional contraction due to the influx of calcium through Ca_V_1.2, receptor- (ROCCs) and store-operated (SOCCs) calcium channels [28]. Several studies have demonstrated the participation of intracellular stores and extracellular calcium on the vascular response to indole-alkaloids. Indole-alkaloids, like rhynchophylline and isorhynchophylline from *Uncaria rhynchophylla* (Miq.) Miq. ex Havil, have been reported to exert a hypotensive and vasodilator effect on isolated thoracic aorta segments through the inhibition of intracellular calcium release [29]. Naucline, derived from *Nauclea officinalis* (Pierre ex Pit.) Merr. and Chun, cadamine, derived from *Ochreinauclea maingayii* (Hook.f.) Ridsdale, and *N*-demethylglypetelotine (glypetelotine), derived from *Glycosmis petelotii* Guillaumin, cause independent endothelium relaxation through the blocking of the Ca_V_1.2 channels in rat aorta [15,30]. Therefore, ARI, an isolated indole-alkaloid from *A. chilensis,* may provide its vasodilation properties through a modulation of calcium release and influx of calcium from extracellular sources through Ca_V_1.2.

The vasodilator proprieties of ARI observed previously could cause a diminution in peripheral resistance. The conductance vasculature (aorta and large vessels) would not be directly involved in the regulation of the blood pressure. Nevertheless, it is expectable that the aristoteline could modulate the myogenic tone of smaller vessels and, by this, help to diminish peripheral resistance.

A limitation of this study is the ex vivo model. It is necessary to note that an ex vivo model of isolated aortic rings is an useful system for testing the vasoreactivity potency of different compounds [31], but does not allow comprehensive evaluation of the potential pharmacological activities. In the terms of the future therapeutic applications of various compounds, the potencies can be achieved only by using the respective in vivo models.

The knowledge of the vasodilator proprieties of ARI would be used in the development of the prototype drug that aims to improve pharmacological activity, such as hypertension, hypoxic pulmonary vasoconstriction, and optimize the pharmacokinetic profile through a characterization and concentration of the medicinal principle.

In further studies, it would be interesting to investigate in vivo the effect of ARI on blood pressure in normotensive and hypertensive animals. Especially important in Bolivia, Perú, and Chile are the studies on the models of long- and short-term chronic intermittent hypoxia. Such information can be used to relate to its effectiveness against acute mountain sickness prevalent in the region.

## 4. Materials and Methods

### 4.1. Drugs

The drugs used were l-phenylephrine hydrochloride (PE), 1*H*-(1,2,4)oxadiazolo[4,3-*a*]quinoxalin-1-one (ODQ), *N*^ω^-nitro-l-arginine methyl ester (l-NAME), and (±)-Bay K8644 (Sigma−Aldrich, St Luis, MO, USA). Caffeine, Tetraethylammonium (TEA), Barium chloride dihydrate (BaCl_2_), 4-aminopyridine (4-AP) and indomethacin were obtained from Merck (Darmstadt, Germany). Except indomethacin, the drugs were dissolved in distilled deionized water (deionized water Millipore) and kept at 4 °C. The stock solution of ARI, indomethacin, and (±)-Bay K8644 were dissolved in dimethyl sulfoxide (DMSO) (Merck, Germany). ARI had purity higher than 98% (by HPLC with MS detector, NMR and melting point measurements). A stock solution in DMSO (10^−1^ M) was prepared with ARI and, from this stock, new dilutions of ARI were made in Krebs–Ringer bicarbonate buffer (KRB) and added into the organ bath. Final DMSO concentration in the organ bath was lower than 0.1%. The Krebs−Ringer bicarbonate (KRB) solutions were freshly prepared before each experiment. Physiological KRB containing (×10^−3^ M): 4.2 KCl, 1.19 KH_2_PO_4_, 120 NaCl, 25 NaHCO_3_, 1.2 MgSO_4_, 1.3 CaCl_2_, and 5 D-glucose (pH 7.4).

### 4.2. Isolation of ARI

Fresh leaves of *A. chilensis* (7 kg) were collected at S 36°50’01.51’’ W 73°01’53.75’’ in Concepción, Chile in August 2014. The compound was extracted from the leaves by placing the leaves in acid water (pH 3, HCl) for three days at room temperature and then filtering the resulting liquid. The aqueous layer was alkalinized to pH 11 with NaOH and subsequently extracted with EtOAc (3×1L). The organic layer was evaporated at 45 °C and 200 mmHg in order to produce a gummy, red residue. The resulting OH^-^ extract was chromatographed using a silica gel column (200−300 mesh) and elevated solvent polarity (from hexane 100% to EtOAc 100%). From that portion, at 60% hexane/EtOAc, a colorless precipitate was obtained and then recrystallized from EtOAc at 4 °C, producing ARI (60 mg, colorless crystals).

The ^1^H and ^13^C-NMR (nuclear magnetic resonance) spectra were recorded in CDCl_3_ on a Bruker Avance III spectrometer (Bruker Biospin GmbH, Rheinstetten, Germany) at 600.13 (^1^H) and 150.61 (^13^C) MHz, with the deuterium signal of the solvent as the lock and TMS (for ^1^H) or the solvent (for ^13^C) as the internal standard. These data and methodology are consistent with previous reports in the literature [16].

### 4.3. Animals

20 Male Sprague–Dawley rats (6–8 weeks of age, 160–190 g) were obtained for vascular reactivity experiments from the breeding colony at Antofagasta University and used in this study. All animals were housed in a temperature-controlled (21 ± 1 °C), light-cycled (08:00–20:00 hours) room with ad libitum access to drinking water and standard rat food (Champion, Santiago). The research methods conformed to the Guide for the Care and Use of Laboratory Animals published by the U.S. National Institutes of Health (NIH Publication, revised 2013). The local animal research committee also approved the experimental procedure that was used in this study (number CEIC 135/2018).

### 4.4. Isolation of Aortic Rings

Rats were sacrificed by cervical dislocation. The thoracic aorta was quickly removed and placed in a cold (4 °C) Krebs–Ringer physiological bicarbonate buffer (KRB). Rings (3–5 mm and 2–4 mg) were prepared after connective tissue was cleaned out from the aorta, taking special care to avoid endothelial damage. Aortic rings were equilibrated for 40 min in KRB at 37 °C by constant bubbling with 95% O_2_ and 5% CO_2_.

### 4.5. Vascular Reactivity Experiments

Aortic rings from the same animal were studied in duplicate, using different vasoactive substances (PE and KCl). The rings were mounted on two 25-gauge stainless steel wires; the lower wires were attached to a stationary glass rod and the upper one was attached to an isometric transducer (Radnoti, Monrovia, California). The transducer was connected to a PowerLab 8/35 (Colorado Springs CO) for the continuous recording of vascular tension using the Lab-Chart Prov 8.1.2 computer program (ADInstrument, Dunedin, New Zealand). After an equilibration period of 30 min, the aortic rings were stabilized by three successive, near-maximum contractions with KCl (60 mM) for 10 min. The passive tension on the aorta was 1.0 g, which was determined to be the resting tension for obtaining maximum active tension induced by the 60 mM concentration of KCl [32]. Each original register was checked, and the aortic rings used did not present significant elevation on the basal tone by pre-incubation with l-NAME (10^-4^ M), potassium blockers (TEA, BaCl_2_, glibenclamide; 10^-5^ M), ODQ (10^-6^ M), indomethacine, or Caffeine (10 mM) before to vascular contractile response to KCl or PE.

Since ARI-induced vasorelaxation may change with pre-contraction level induced by phenylephrine or KCl, we elicited matching contractions with phenylephrine or KCl with different protocols used before to ARI-induced vasorelaxation.

### 4.6. Assessment of the Effect of ARI on the Vasodilation in Isolated Aortic Rings Pre-Contracted with PE, with and without Endothelium

In the first protocol, the aortic rings were pre-contracted with 1 μM PE and then increasing concentrations of ARI (1 nM to 100 μM) were added to the bath. The endothelium removal was performed by gently rubbing off the endothelium using a small piece of cotton. The vasodilation to 10 μM acetylcholine (muscarinic agonist) in pre-contracted aortic rings with 1 μM PE was used to evaluate the endothelium vascular function. The aortic rings were considered to have shown a functional endothelial response if vasodilation increased by 70–80% [31].

In the second protocol, the role of endothelial nitric oxide on rat aorta was studied. An inhibitor of endothelial nitric oxide synthase (l-NAME) and a highly selective, irreversible, heme-site inhibitor of soluble guanylyl cyclase (ODQ) were used. The aortic rings were pre-incubated with l-NAME (100 μM) or ODQ (1 μM) for 20 min before the experiment. Then, the aortic rings were pre-contracted with 1 μM PE, and then increasing concentrations ARI (1 nM to 100 μM) were added to the bath. We elicited matching contractions with PE in the presence or absence of l-NAME and ODQ before recording the ARI-induced relaxation.

### 4.7. Assessment of the Effect of ARI on the Contractile Response to KCl and PE

In this experiment, the effect of ARI on contractile response to KCl or PE was analyzed. Intact rat aortic rings were pre-incubated in the presence of ARI (10 μM) for 20 min, followed by the addition of KCl (10 mM to 60 mM) or PE (0.1 nM to 10 μM).

### 4.8. Assessment of the Role of Intracellular and Extracellular Calcium in the Vascular Response to ARI

To evaluate the role of extracellular calcium, experiments were performed with a calcium-free KRB containing (mM): 1.0 ethylene glycol tetraacetic acid (EGTA), 4.2 KCl, 1.19 KH_2_PO_4_, 125 NaCl, 25 NaHCO_3_, 1.2 MgSO_4_, and 5 D-glucose (pH 7.4). The aortic rings were pre-incubated in a KRB with calcium for 30 min. Then, the KRB was changed with a calcium-free KRB for 10 min before PE (10^−6^ M) was added. Five minutes after contraction was induced with PE, cumulative concentrations of CaCl_2_ (0.1 to 1 mM) were added to the medium [33,34]. In other experiments, BaCl_2_ or TEA was used to stimulate the vasoconstriction via barium influx in L-type calcium channels and the blocking of inward rectifying potassium channels, thus depolarizing the plasma membrane. The contraction was induced by 1 mM BaCl_2_ or 1 mM TEA for 10 min. The aortic rings were pre-incubated with ARI (10 μM) for 20 min before the experiment.

### 4.9. Assessment of the Role of Potassium Channels in the Vascular Response to ARI

To study the role of potassium channels on the vascular response of rat aorta, TEA, BaCl_2_, and glibenclamide were used. The aortic rings were pre-incubated with TEA (10 μM ), BaCl_2_ (10 μM ) or glibenclamide (10 μM ) for 20 min before the experiment. Then, the aortic rings were pre-contracted with 1 μM PE and then increasing concentrations of ARI (1 nM to 100 μM) were added to the bath. The pre-incubation of aortic rings with low concentrations of different potassium channel blockers, such as TEA (10 μM), BaCl_2_ (10 μM), and glibenclamide (10 μM) did not elevate per se the basal vascular tone, nor did the amplitude of pre-contraction with phenylephrine (1 µM).

### 4.10. Statistical Analysis

Values were expressed as mean ± standard error (S.E.M.). A repeated-measures two-way analysis of variance (ANOVA) was carried out to detect significant differences between dose-response curves, followed by a Bonferroni post hoc test. Area under the curve (AUC) was calculated using the trapezoidal rule. In addition, the determination of IC_50_ was performed using nonlinear regression (sigmoidal) via Graph Pad PrismTM software, version 5.0. (GraphPad Software, Inc., La Jolla, CA, USA). Paired and unpaired Student’s *t*-tests were also used when appropriated. The plot and linear regression was performed by using the least square method. A P-value of <0.05 was considered statistically significant.

## 5. Conclusions

This study demonstrates that the natural alkaloid (ARI) obtained from of *A. chilensis*, a tree employed in Chilean traditional medicine, is able to reduce the tone of arterial smooth muscle. The vasodilator effect of ARI involves responses independent of endothelium, probably due to calcium channels blockage and/or activation of potassium channels. These findings have potential clinical effect for the therapy of hypoxic episodes, such as ischemia or acute mountain sickness, because the block calcium influx and the activation of the potassium channels could cause a reduction in myocardial oxygen demand, in the same way as do calcium blockers [35]. Results from these investigations have clearly shown that there is a strong relationship between the ethnopharmacological uses of *Aristotelia* and its pharmacological effects [1,4].

## Figures and Tables

**Figure 1 molecules-24-02748-f001:**
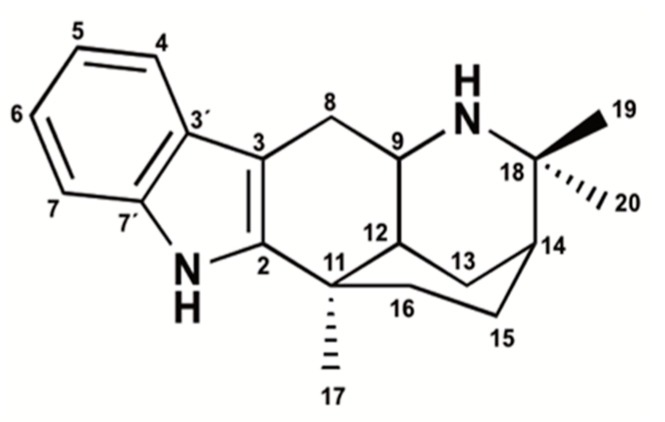
Chemical structure of aristoteline (ARI).

**Figure 2 molecules-24-02748-f002:**
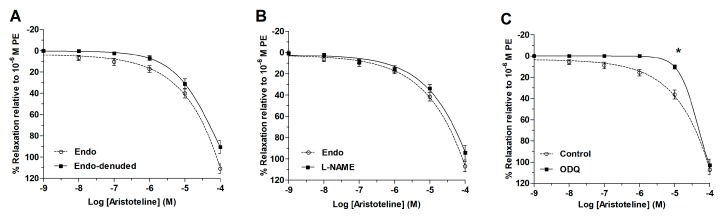
ARI directly causes relaxation on vascular smooth muscle of aorta. Vasodilatation response induced by ARI (1 nM to 100 μM) after the addition of phenylephrine (PE) (1 μM) is shown in intact aortic rings (endo or control) and denuded aorta (endo-denuded) (**A**), preincubate with L-NAME (100 μM) (**B**) or with 1*H*-(1,2,4)oxadiazolo[4,3-*a*]quinoxalin-1-one (ODQ) (1 μM) (**C**). Values are mean ± standard error of the mean of 6 experiments. The determination of IC_50_ were performed using nonlinear regression and a repeated-measures two-way analysis of variance (ANOVA) to compare curves. Statistically significant difference **p* < 0.05 vs. Control.

**Figure 3 molecules-24-02748-f003:**
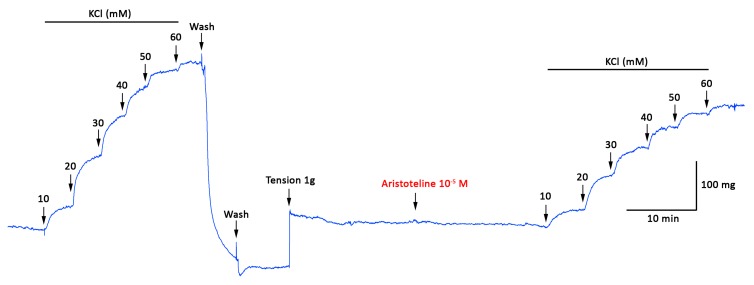
Original trace showing the time course of the concentration–response curves to KCl in intact aortic ring from rat in absence and presence of ARI (10^−5^ M). The aortic ring was pre-incubated with ARI for 20 min before KCl was added.

**Figure 4 molecules-24-02748-f004:**
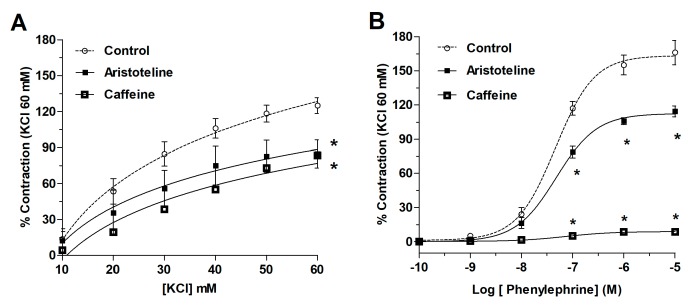
Effect of ARI on the contractile response to KCl (**A**) and PE (**B**) in rat aortic ring. The aorta was pre-incubated in absence (control) or presence of ARI (10 μM) or caffeine (10 mM) for 20 min. Values are the mean ± standard error of the means of 6 experiments. The asterisks in (**A**) refer to both, ARI and caffeine. The determination of EC_50_ were performed using nonlinear regression and a repeated-measures two-way ANOVA to compare curves. Statistically significant difference **p* < 0.05 vs. Control.

**Figure 5 molecules-24-02748-f005:**
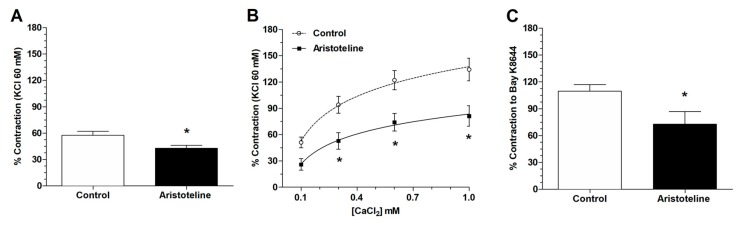
Effect of ARI on calcium release from the phenylephrine-sensitive intracellular calcium stores. The aortic rings were pre-incubated in a free calcium buffer for 10 min before PE was added (**A**), and then, the CaCl_2_ (0.1, 0.3, 0.6, and 1.0 mM) was added to the bath (**B**). Vasoconstriction occurred just when the agonist of Ca_V_1.2 channels (10 nM Bay K8644) was added with 15 mM KCl to the bath (**C**). The aorta was pre-incubated in absence (control) or presence of ARI (10 μM) for 20 min. Values are mean ± standard error of the mean of 6 experiments. Statistically significant difference **p* < 0.05 vs. Control.

**Figure 6 molecules-24-02748-f006:**
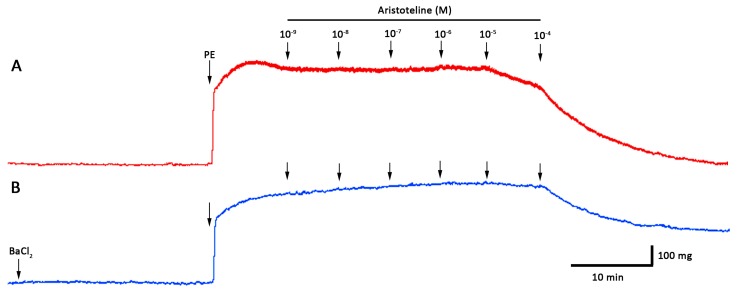
Original trace showing the time course of the concentration–response curves to ARI in intact aortic ring from rat in absence (**A**) and presence of BaCl_2_ (10 μM; **B**). The aortic ring was pre-incubated with BaCl_2_ for 20 min before PE (1 μM) was added.

**Figure 7 molecules-24-02748-f007:**
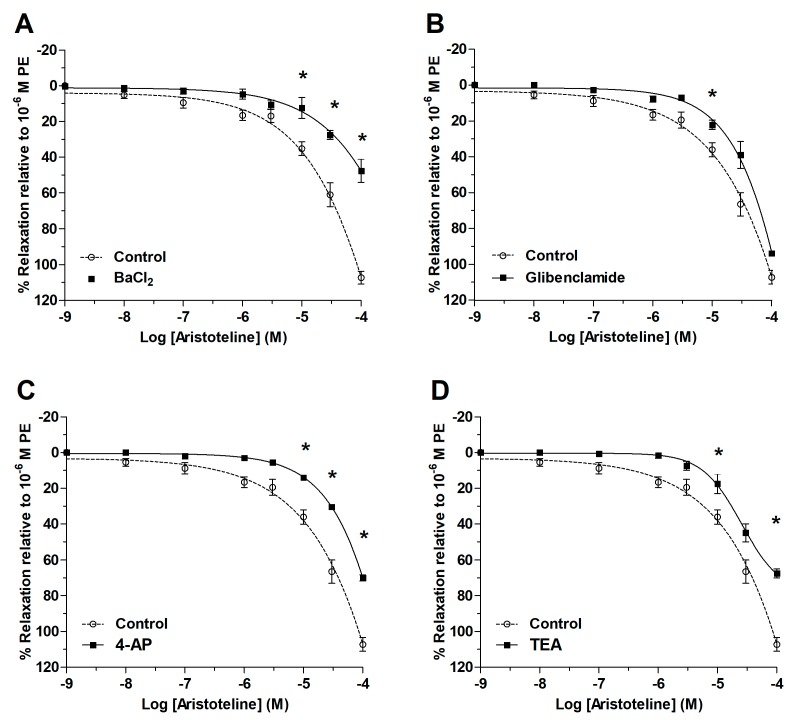
Evaluation of the ARI vasodilatation mechanism associated with potassium channels. Effect of ARI after the addition of BaCl_2_ (10 μM) as a blocker of inward rectifier potassium channels (Kir) is shown (**A**), glibenclamide (10 μM) as blocker of adenosine triphosphate (ATP)-sensitive K^+^ channels (**B**), 4-AP (1 mM) as a non-selective blocker of Kv channels (**C**) and tetraethylammonium (TEA) (1 mM) as a blocker of KCa channels (**D**). PE (1 μM) was used to induce the contractile responses to the aortic rings. The determination of IC_50_ were performed using non-linear regression and a repeated-measures two-way ANOVA to compare curves; *n* = 6. Statistically significant difference **p* < 0.05 vs. Control.

**Figure 8 molecules-24-02748-f008:**
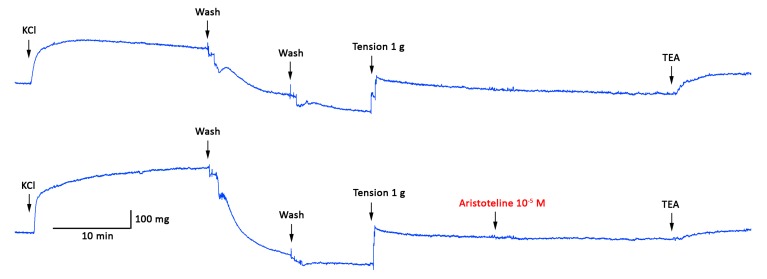
Original tracing showing the time course of the contractile response to KCl (60 mM) and TEA (1 mM) in intact aortic rings from rats. These are in the absence or presence of ARI (10^-5^ M). The aortic rings were pre-incubated with ARI for 20 min before TEA was added.

**Figure 9 molecules-24-02748-f009:**
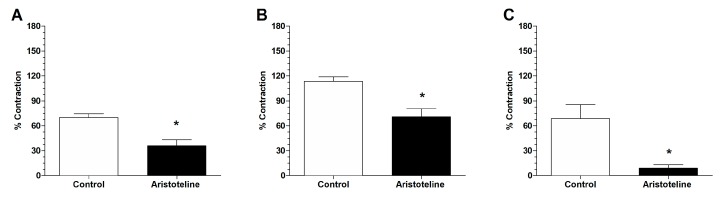
Evaluation of the ARI contraction mechanism associated with potassium channels. The aortic rings were pre-contracted with KCl (15 mM) (**A**), BaCl_2_ (1 mM) as a blocker of inward rectifier potassium channels (Kir) (**B**), TEA (1 mM) as a non-selective blocker of KCa1.1 channels (**C**). The vascular tissue was pre-incubated in absence (control) or presence of ARI (10 μM) for 20 min. Unpaired Student’s t-tests; *n* = 6. Statistically significant difference **p* < 0.05 vs. Control.

**Figure 10 molecules-24-02748-f010:**
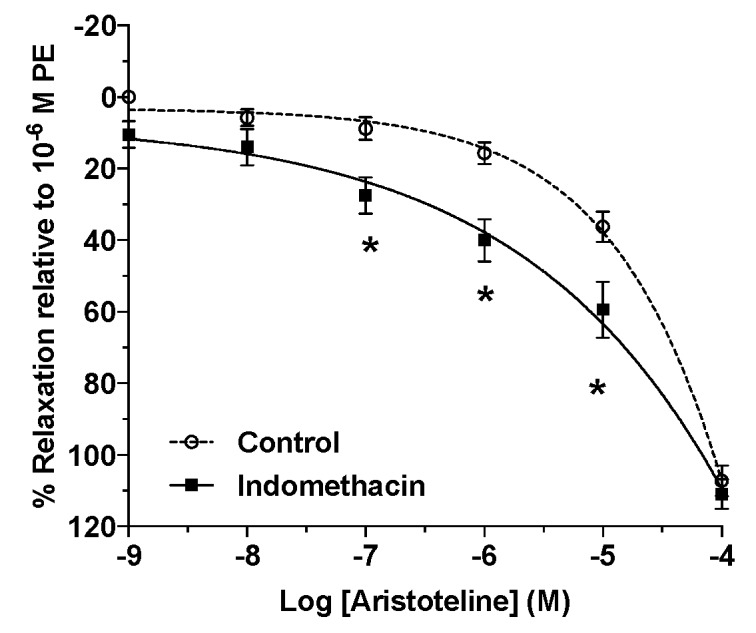
Evaluation of the ARI vasodilatation mechanism associated with endothelial prostanoids. Data shows the effect of ARI after the addition of indomethacin (10 μM) as a non-selective cyclooxygenase inhibitor. PE (1 μM) was used to induce the contractile responses. The determination of IC_50_ were performed using nonlinear regression and a repeated-measures two-way ANOVA to compare curves; *n* = 6. Statistically significant difference **p* < 0.05 vs. Control.

**Table 1 molecules-24-02748-t001:** Effect of ARI on the vascular response to different vasoactive substances in rat aorta.

Drugs	IC_50_ (μM)	AUC (× 10^−3^)
Control	15.86 ± 0.91	7.17 ± 0.31
Endo-denuded	14.96 ± 0.89	5.63 ± 0.49
l-NAME	14.39 ± 0.89	6.37 ± 0.25
ODQ	39.75 ± 0.87 *	5.14 ± 0.24 *
BaCl_2_	19.64 ± 0.83 *	2.52 ± 0.52 *
Glibenclamide	28.79 ± 0.89 *	5.34 ± 0.17 *
TEA	18.65 ± 0.91 *	5.62 ± 0.25
4-AP	29.32 ± 0.93 *	4.04 ± 0.05 *
Indomethacin	5.20 ± 0.72 *	7.76 ± 0.10

IC_50_ and AUC represent the half maximal inhibitory concentration and area under curve, respectively. The values are mean ± standard error of the mean (S.E.M.) and represents the mean of at 6 independent experiments. Statistically significant difference * *p* < 0.05 vs. Control.

**Table 2 molecules-24-02748-t002:** Effect of ARI on the vascular response to KCl, phenylephrine (PE) and calcium in rat aorta.

Drugs	EC_50_	AUC
**KCl (mM)**		
Control	23.0 ± 1.3	4482 ± 163
ARI	26.0 ± 0.8 *	3212 ± 293 *
Caffeine	29.0 ± 2.1*	2300 ± 5 *
**PE (nM)**		
Control	47.8 ± 0.2	1.56 ± 0.08 (×10^−3^)
ARI	51.2 ± 0.1 *	1.01 ± 0.03 * (×10^−3^)
Caffeine	59.56 ± 0.63 *	0.08 ± 0.02 * (×10^−3^)
**Calcium (mM)**		
Control	0.30 ± 0.12	89.64 ± 8.54
ARI	0.31 ± 0.17	47.03 ± 7.09 *

EC_50_ represents the half maximal effective concentration. The values are mean ± S.E.M. represents the mean of at 6 independent experiments. Statistically significant difference * *p* < 0.05 vs. Control.

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
