# Peer review of "Aristoteline, an Indole-Alkaloid, Induces Relaxation by Activating Potassium Channels and Blocking Calcium Channels in Isolated Rat Aorta"

_molecules, 2019, doi:10.3390/molecules24152748_

Reviewer 2 Report

This manuscript by Romero et al. investigated that the effects of indole alkaloid aristoteline (ARI) and vascular tone regulation in rat aorta. Using organ bath technique, the authors found that the compound lead to relaxation via endothelium-independent mechanisms. Moreover, the authors found that such relaxations induced by the compound were mediated by K+ channel opening, a calcium-induced antagonism, and possibly activation of sGC. Overall, the data are interesting. However, several concerns should be addressed.

 Concerns

1. Firstly, the manuscript, as a whole, the language and abbreviation needs to be checked again. For example, please revise from Bay K6844 to Bay K8644. Lines 71 and 72, please check bold font of references

2. A major concern raised by the reviewer is data analysis. Did the authors use repeated measures ANOVA?  

3. Although the authors found several interesting data of the compound, some important data were lacking in this manuscript. What were the levels of NO or its metabolites, cGMP, phosphorylated eNOS in this experiment? Since high concentration of TEA is non-specific inhibitor, for example, the authors should investigate the effects of apamin, TRAM-34, or iberiotoxin on the compound-induced relaxation. Moreover, the authors should investigate not only large artery but also resistance size artery.

4. In all concentration-response curve, the authors should show representative trace. Moreover, please show time course experiment by using control (vehicle; DMSO) because the relaxation (especially final application of this compound) may be just tone down of precontraction.

5. In text, if the author use “significantly”, please state comparison group.

6. Please insert n number in all figure legends.  

7. I think that -LogIC50 (or -LogEC50) and IC50 (or EC50) are similar pharmacological information, thus, please delete one. Moreover, it may be more informative showing AUC.

8. The authors should measure cGMP levels in this study. Moreover, if the authors can measure membrane potential and intracellular calcium levels, these data may be strongly supporting the author’s hypothesis. Please address them.

9. In Discussion, the authors should discuss what is novelty and pathophysiological importance in these findings more details. Especially, the authors used large artery but not resistance size artery. If the authors only discuss about regulation of blood pressure (i.e., anti-hypertensive effect), please discuss more details based on these points.

10. I do not believe some drugs could dissolve water (e.g., indomethacin). Please explain it. Did the authors use just KCl solution or high-K+ solution?

11. Please make a schematic summary figure.

Author Response

Reviewer 2

Comments and Suggestions for Authors

This manuscript by Romero et al. investigated that the effects of indole alkaloid aristoteline (ARI) and vascular tone regulation in rat aorta. Using organ bath technique, the authors found that the compound lead to relaxation via endothelium-independent mechanisms. Moreover, the authors found that such relaxations induced by the compound were mediated by K+ channel opening, a calcium-induced antagonism, and possibly activation of sGC. Overall, the data are interesting. However, several concerns should be addressed.

 Concerns

1. Firstly, the manuscript, as a whole, the language and abbreviation needs to be checked again. For example, please revise from Bay K6844 to Bay K8644. Lines 71 and 72, please check bold font of references. We have corrected them.

 2. A major concern raised by the reviewer is data analysis. Did the authors use repeated measures ANOVA?  Yes, a repeated-measures two-way analysis of variance (ANOVA) was carried out to detect significant differences between dose-response curves, followed by a Bonferroni post hoc test. We included this statement in the Statistical analysis section and the figure legends.

3. Although the authors found several interesting data of the compound, some important data were lacking in this manuscript. What were the levels of NO or its metabolites, cGMP, phosphorylated eNOS in this experiment? Since high concentration of TEA is non-specific inhibitor, for example, the authors should investigate the effects of apamin, TRAM-34, or iberiotoxin on the compound-induced relaxation. Moreover, the authors should investigate not only large artery but also resistance size artery.

Since the results with L-NAME and endothelium-denuded rat aorta showed that endothelial nitric oxide is not involved in the relaxation by aristoteline, it is very probably that determination of the levels of NO or phosphorylated eNOS will not provide relevant information. We realize that new experiments are needed to analyze in depth the role of cGMP-pathway, but we believe that it is not the novelty of this study, therefore we have eliminated some statements about this subject.

We agree with reviewer that TEA is relative non-specific blocker of calcium channels. It blocks large conductance calcium-activated potassium currents (KCa1.1) [1] and voltage-dependent potassium channels, such as Kv1, Kv2 and Kv3 [2]. Apamin is a peptide neurotoxin that blocks small conductance calcium-activated potassium channels, such as KCa2.1-2.3 isoforms [3]. While Iberiotoxin is a selective blocker of high-conductance calcium-activated potassium channels, such as KCa1.1 [3]. At last, TRAM-34 is a potent selective KCa3.1 channel blocker [4]. On the other hand, the calcium-activated potassium channels that participate in the regulation of contraction of the vascular smooth muscle in rat aorta are of the large (BKCa) and intermediate conductance (KCa3.1) [5].  We have corrected the statement about TEA in the Discussion section.

Although we did not investigate resistance arteries, rat aorta assay provides useful information about the mechanisms of vasoconstriction and relaxation, and is a good model between in vivo and in vitro model studies [6].

 4. In all concentration-response curve, the authors should show representative trace. Moreover, please show time course experiment by using control (vehicle; DMSO) because the relaxation (especially final application of this compound) may be just tone down of precontraction.

We have included several representative traces of concentration-response curve in the manuscript (Fig. 3, Fig. 6 and Fig. 8).

This point was raised by the 1st reviewer in earlier comments, and we provided the information in the Drugs section: “A stock solution in DMSO (10−1 M) was prepared with ARI and, from this stock, new dilutions of ARI were made in Krebs-Ringer bicarbonate buffer (KRB) and added into the organ bath. Final DMSO concentration in the organ bath was lower than 0.1%.” Please find in Fig. 1 the effect of DMSO (0.00001% to 0.1%). These concentrations of DMSO were the same in presence of ARI (ARI 10-8 to 10-4 M).

Figure 1: Original trace showing the time course of the concentration–response curves to DMSO (without ARI 10-8 to 10-4 M) in intact aortic ring from rat.

 5. In text, if the author use “significantly”, please state comparison group. We have rephrased the statements to reflect the significant differences, and the groups concerned.

 6. Please insert n number in all figure legends.  We added “n= 6” in Figures 6, 8 and 9. In the others figures, “Values are mean ± standard error of the mean of 6 experiments” was used to reflect the n values

 7. I think that -LogIC50 (or -LogEC50) and IC50 (or EC50) are similar pharmacological information, thus, please delete one. Moreover, it may be more informative showing AUC. We have deleted the EC50 and we have calculated the AUC in Table 1 and Table 2.

 8. The authors should measure cGMP levels in this study. Moreover, if the authors can measure membrane potential and intracellular calcium levels, these data may be strongly supporting the author’s hypothesis. Please address them.

We agree with the reviewer that measurement of cGMP levels will provide new information to support our hypothesis in the mechanisms of aristoteline. We hope to assess and use these methodologies for further studies.

The vasodilation properties of aristoteline through a modulation of influx of calcium from extracellular sources mediate by CaV1.2 were confirmed by free calcium and Bay K8644 experiments, which activates specifically voltage-sensitive L-type calcium channels (Cav1.2) [7,8].

 9. In Discussion, the authors should discuss what is novelty and pathophysiological importance in these findings more details. Especially, the authors used large artery but not resistance size artery. If the authors only discuss about regulation of blood pressure (i.e., anti-hypertensive effect), please discuss more details based on these points.

These findings have potential clinical effect for the therapy of hypoxic episodes, such as ischemia or acute mountain sickness, because the block the calcium influx, through the activation of the potassium channels. This could lead to a reduction in myocardial oxygen demand, in the same way as do calcium blockers [9]. We have included it in the Conclusions section.

The vasodilator proprieties of ARI observed in this study could cause a decrease in peripheral resistance even in smaller resistance vessels. The conductance vasculature (aorta and large vessels) would not be directly involved in the regulation of the blood pressure. Nevertheless, it is expectable that the aristoteline could modulate the myogenic tone of smaller vessels and, by this, help to diminish peripheral resistance. Please find similar response in the point 3 above. We have included an explanation in the Discussion section.

 10. I do not believe some drugs could dissolve water (e.g., indomethacin). Please explain it. Did the authors use just KCl solution or high-K+ solution? We agree with the solution of indomethacin, which dissolved in DMSO. We have included the information in the Drugs section. We added KCl 150 mL of KCl 4 M for obtained a final concentration of 60 mM in the bath (10 mL).

 11. Please make a schematic summary figure. Please find the schematic summary figure in the graphical abstract.

References

1.            Gutman, G.A.; Chandy, K.G.; Grissmer, S.; Lazdunski, M.; McKinnon, D.; Pardo, L.A.; Robertson, G.A.; Rudy, B.; Sanguinetti, M.C.; Stühmer, W., et al. International Union of Pharmacology. LIII. Nomenclature and molecular relationships of voltage-gated potassium channels. Pharmacol Rev 2005, 57, 473-508, doi:10.1124/pr.57.4.10.

2.            Li, S.; Choi, V.; Tzounopoulos, T. Pathogenic plasticity of Kv7.2/3 channel activity is essential for the induction of tinnitus. Proc Natl Acad Sci U S A 2013, 110, 9980-9985, doi:10.1073/pnas.1302770110.

3.            Wang, M.; Xing, N.; Wu, L.; Huang, W.C.; Xu, Z.; Liu, G. Regulation of Spontaneous Contractions in Intact Rat Bladder Strips and the Effects of Hydrogen Peroxide. Biomed Res Int 2018, 2018, 2925985, doi:10.1155/2018/2925985.

4.            D'Alessandro, G.; Catalano, M.; Sciaccaluga, M.; Chece, G.; Cipriani, R.; Rosito, M.; Grimaldi, A.; Lauro, C.; Cantore, G.; Santoro, A., et al. KCa3.1 channels are involved in the infiltrative behavior of glioblastoma in vivo. Cell Death Dis 2013, 4, e773, doi:10.1038/cddis.2013.279.

5.            Jackson, W.F. Potassium Channels in Regulation of Vascular Smooth Muscle Contraction and Growth. Adv Pharmacol 2017, 78, 89-144, doi:10.1016/bs.apha.2016.07.001.

6.            Rameshrad, M.; Babaei, H.; Azarmi, Y.; Fouladia, D.F. Rat aorta as a pharmacological tool for in vitro and in vivo studies. Life Sci 2016, 145, 190-204, doi:10.1016/j.lfs.2015.12.043.

7.            Kanmura, Y.; Itoh, T.; Kuriyama, H. Agonist actions of Bay K 8644, a dihydropyridine derivative, on the voltage-dependent calcium influx in smooth muscle cells of the rabbit mesenteric artery. J Pharmacol Exp Ther 1984, 231, 717-723.

8.            Sarkar, S.; Ravikumar, B.; Floto, R.A.; Rubinsztein, D.C. Rapamycin and mTOR-independent autophagy inducers ameliorate toxicity of polyglutamine-expanded huntingtin and related proteinopathies. Cell Death Differ 2009, 16, 46-56, doi:10.1038/cdd.2008.110.

9.            Hackett, P.H.; Roach, R.C. High-altitude illness. N Engl J Med 2001, 345, 107-114, doi:10.1056/NEJM200107123450206.

Reviewer 3 Report

The manuscript by Romero et al. describes the effect of the alkaloid aristoteline (ARI) on the vascular reactivity using an ex-vivo system. The methods are well described and the experiments performed are clear. Unfortunately, the style of the manuscript is hard to follow. The introduction could be improved, whilst Discussion partly repeats results, does not interpret results sufficiently and I found some speculations of the results. Grammar also needs moderate improvement. Although the authors draw the right conclusions there is work to do on the manuscript.

- The authors affirm that 4-AP (1 mM) is a non-selective blocker of Kv channels. I would like to know what type of Kv channels are the authors referring to? The family of Voltage gated potassium channels (Kv) is composed by a vast number of subtypes.

- Are you sure that Tetraethylammonium (TEA) is a good and selective blocker of KCa1.1 channels? TEA blocks other Ca2+-activated K+ channels, several Kv channels, it is an inhibitor of nicotinic acetylcholine receptors and inhibits the Na,K-ATPase.

- Lines 52-69: These sentences could be deleted or moved up

- Line 88: In order to induce tonic contraction, the aortic rings, with and without endothelium, were stimulated with PE (1 uM)

- Line 103: Please use synonyms for “decrease”, “increase” and “significantly” in order to avoid repetitions

- Line 184: The results obtained suggest

- Line 185-188: This sentence is hard to follow. It would be appropriate to re-formulate these sentences because they are not properly understood

.- Line 199: The “gating (or activation)” instead of opening

- Line 261: Please define here the composition of KRB buffer

-Please check all the abbreviations and symbols. Introduce the abbreviation (for example Aristoteline (ARI)), and then always use the same abbreviation. For Instance, I found “calcium”, then “Ca2+”, the again “calcium”.

Minor points:

- Line 45: ARI should be “Aristoteline (ARI)”

- Line 54: It has been hypothesized….

- Line 76: Thus, the aim of this study has been the evaluation of the effect….

- Line 138: agonist calcium channel ?

Author Response

Reviewer 3

Comments and Suggestions for Authors

The manuscript by Romero et al. describes the effect of the alkaloid aristoteline (ARI) on the vascular reactivity using an ex-vivo system. The methods are well described and the experiments performed are clear. Unfortunately, the style of the manuscript is hard to follow. The introduction could be improved, whilst Discussion partly repeats results, does not interpret results sufficiently and I found some speculations of the results. Grammar also needs moderate improvement. Although the authors draw the right conclusions there is work to do on the manuscript.

 - The authors affirm that 4-AP (1 mM) is a non-selective blocker of Kv channels. I would like to know what type of Kv channels are the authors referring to? The family of Voltage gated potassium channels (Kv) is composed by a vast number of subtypes. 4-AP has greatest potency at Kv1 and Kv3 family members at micromolar concentrations, and also blocks Kv2 and Kv4 at millimolar concentrations [1]. We have indicated in the manuscript the type of Kv channels.

 - Are you sure that Tetraethylammonium (TEA) is a good and selective blocker of KCa1.1 channels? TEA blocks other Ca2+-activated K+ channels, several Kv channels, it is an inhibitor of nicotinic acetylcholine receptors and inhibits the Na,K-ATPase. We agree with the reviewer that TEA is relative non-specific blocker of calcium channels. It blocks Calcium-activated potassium currents (KCa1.1) [1] and voltage-dependent potassium channels, such as Kv1, Kv2 and Kv3 [2]. We have corrected this statement.

 - Lines 52-69: These sentences could be deleted or moved up. We have deleted a few paragraphs and reordered some sentences of the Introduction.

 - Line 88: In order to induce tonic contraction, the aortic rings, with and without endothelium, were stimulated with PE (1 uM). We have deleted this sentence and rewritten this idea.

- Line 103: Please use synonyms for “decrease”, “increase” and “significantly” in order to avoid repetitions. We have used synonyms for “decrease” and “increase”, but “significantly” has a specific statistical meaning.

- Line 184: The results obtained suggest. We have included this words in the beginning of the Discussion section.

- Line 185-188: This sentence is hard to follow. It would be appropriate to re-formulate these sentences because they are not properly understood. We have rewritten this sentence.

 .- Line 199: The “gating (or activation)” instead of opening. We have changed “opening” to “activation”.

 - Line 261: Please define here the composition of KRB buffer. We have included the composition of the KRB in Drugs section.  

-Please check all the abbreviations and symbols. Introduce the abbreviation (for example Aristoteline (ARI)), and then always use the same abbreviation. For Instance, I found “calcium”, then “Ca2+”, the again “calcium”. We have corrected the abbreviations and symbols in the text.

Minor points:

 - Line 45: ARI should be “Aristoteline (ARI)”. We have corrected it.

- Line 54: It has been hypothesized…. We have corrected it.

- Line 76: Thus, the aim of this study has been the evaluation of the effect…. We have corrected it.

- Line 138: agonist calcium channel? We have completed this sentence: “… of CaV1.2 channels…”

References

1.            Gutman, G.A.; Chandy, K.G.; Grissmer, S.; Lazdunski, M.; McKinnon, D.; Pardo, L.A.; Robertson, G.A.; Rudy, B.; Sanguinetti, M.C.; Stühmer, W., et al. International Union of Pharmacology. LIII. Nomenclature and molecular relationships of voltage-gated potassium channels. Pharmacol Rev 2005, 57, 473-508, doi:10.1124/pr.57.4.10.

2.            Li, S.; Choi, V.; Tzounopoulos, T. Pathogenic plasticity of Kv7.2/3 channel activity is essential for the induction of tinnitus. Proc Natl Acad Sci U S A 2013, 110, 9980-9985, doi:10.1073/pnas.1302770110.

Round  2

Reviewer 2 Report

Although the authors have addressed some of my comments, I could not find original trace showing the time course of control group. Please show it.

Figure 1: Original trace showing the time course of the concentration–response curves to DMSO (without ARI 10-8 to 10-4 M) in intact aortic ring from rat.

Reviewer 3 Report

The authors has improved the quality of the paper, therefore I recommend the manuscript for publication in its present state

This manuscript is a resubmission of an earlier submission. The following is a list of the peer review reports and author responses from that submission.

Molecules, Manuscript ID: molecules-443696

Round  1

Reviewer 1 Report

This study investigates the mechanisms underlying the vasorelaxant effects of aristoteline, an indole alkaloid isolated from Aristotelia chilensis (Molina) Stuntz 26 (Maqui), in in vitro, normotensive rat aorta rings. The data presented demonstrate that aristoteline acts through the opening of potassium channels, inhibition of extracellular calcium influx and activation of the nitric oxide-independent soluble guanylate cyclase. These findings are of limited physiological relevance, since aristoteline causes a significant decrease of vessel tone only at high concentrations. The experiments are logically designed and carefully executed; their presentation, however, can be improved by showing original recordings of, at least, the most important experiments.

Ion channel modulation is claimed on the basis of indirect experimental evidences and, therefore, should be mitigated. Furthermore, additional experiments using more specific pharmacological tools should be performed to finalise the conclusions related to aristoteline-induced effects on cellular Ca2+ homeostasis. I also have some concerns regarding methodology, statistical analysis and interpretation of results. The English style should be carefully revised.

Major Comments

General

1.     The authors assert that “Aristotelia chilensis (Molina) Stuntz 26 (Maqui) has shown important medicinal properties in traditional usage”, that “The fruits and leaves of Aristotelia chilensis (Molina) Stuntz has shown an important range of biological properties including antioxidative, anti-inflammatory and analgesic effects”, etc. These statements should be supported by published literature.

Methods

1.     Several pharmacological tools were used to investigate the targets of aristoteline vasorelaxing activity. As stated in the text, each agent was added to the bath solution several minutes before phenylephrine- or high KCl-induced contraction. First, the effect exerted by each compound on the tone induced by the same concentration of the stimulating agent should be reported. Second, in case the phenylephrine- or KCl-induced tone under the various experimental conditions was different, this might have influenced the subsequent aristoteline-induced vasorelaxation.

2.     The use of 4-amino pyridine as a blocker of KV channels might help the analysis of aristoteline mechanism of action.

3.     I am concerned by the use of the same concentration-response curve to aristoteline recorded under control conditions for six different experiments. Was control repeated in parallel with all the six experimental conditions? If this was not the case, results might be severely influenced by a probable inaccuracy due to biological variability.

4.     Statistical analysis. “Beware of using two-way ANOVA to compare curves or to analyse data where one of the factors is concentration. In fact, the ANOVA calculations ignore the concept of trend” (see Fitting models to biological data. GraphPad Prism).

Results

1.     It is clear that aristoteline relaxes aorta ring muscle tone. However, it is necessary to quantify this effect by showing the pIC50 values recorded under the various experimental conditions. Furthermore, introducing the 3 and 30 µM concentrations will aid the correct calculation of this key parameter.

2.     I suggest to show original recordings of the most significant experiments.

3.     Smooth muscle contraction induced by phenylephrine in the absence of extracellular calcium is due to the release of calcium from the intracellular stores through the synthesis of IP3. Restoring extracellular calcium causes an additional contraction due to the influx of calcium through CaV1.2, receptor- and store-operated calcium channels (McFadzean & Gibson, 2002, Br. J. Pharmacol. 135, 1-13). This should be taken into account when hypothesizing the mechanisms underlining the vasorelaxing activity of aristoteline.

4.     The best protocol to study the effect of a drug on CaV1.2 channel opening-induced contraction is to construct a concentration-response curve to calcium in rings depolarised with high KCl in the presence of 1 mM EGTA. This protocol resembles that based on the use of Bay K 8644 plus 15 mM KCl.

5.     I am concerned by the use of 10 µM TEA as a concentration which selectively blocks large and low conductivity potassium channels. In fact, TEA is commonly employed at 1 mM concentration to selectively block KCa1.1 channels (previously named BKCa channels) and at 10 mM concentration to unspecifically block potassium channels.

Discussion

1.     The discussion is very confusing. Some paragraphs lack a proper conclusion (e.g., l. 160-163, l. 179-183), are too speculative (e.g., l. 168-169: only patch-clamp experiments can substantiate this statement) or need to be rephrased (e.g., l. 188-196).

Minor Comments:

General

1.     Please, replace “voltage-dependent calcium channels type L” with “CaV1.2 channels” throughout the text.

Introduction

1.     P. 2 l. 46-48 This sentence is unclear. Please, rephrase.

2.     P. 2 l. 70-73 These statements are not supported by references. Furthermore, what is “pulse”?

Materials and methods

1.     “*P < 0.05, **P < 0.01 …”. You reasonably set P as<0.05. If you want to show <0.01 or 0.001 in the Figures/Tables then you have to set your threshold P as <0.01 and 0.001, respectively, and declare anything outside (i.e., <0.05) as not significant. Use only one level of P, and stick to it. This is a commonplace error.

2.     Was the Bay K 8644 used a mixture of (+) (calcium antagonist) and (-) (calcium agonist) isomers or the pure (-) isomer? Which solvent was used to dissolve it?

3.     Subheadings need to be extensively revised as they are not clear.

4.     P. 8 l. 270-271. Rat aorta rings are not a pharmacological tool.

Results

1.     The molecular structure of aristoteline would be helpful to the reader.

2.     P. 2 l. 85. Replace “dose” with “concentration”.

3.     Phenylephrine-induced contraction in the absence of extracellular calcium was recorded in the presence of 1 mM EGTA. Was this considered when restoring extracellular calcium? In other words: does the calcium concentration reported in Figure 3B indicate free calcium?

Discussion

1.     P. 6 l. 184. Please, introduce this data in the Result section.

Conclusion

1.     Please, provide a reference to support the statement on l. 312-313.

Figures

1.     I suggest to indicate the statistical test used in all the legends to Figures.

2.     Figure 2A. It is not clear why the response to 60 mM KCl is about 130% of the contraction induced by the same concentration of the same agent (as stated on the Y axis). Furthermore, it is not clear whether the asterisks refer to caffeine, aristoteline, or both.

3.     Legend to Figure 3. Please indicate the aristoteline incubation time.

4.     Legend to Figure 4. Please, indicate the stimulating agent used to contract vessels.

5.     Figure 5. The Y axis titles are not clear. Furthermore, aristoteline incubation time and concentration should be stated in the legend.

Author Response

Reviewer 1

This study investigates the mechanisms underlying the vasorelaxant effects of aristoteline, an indole alkaloid isolated from Aristotelia chilensis (Molina) Stuntz 26 (Maqui), in in vitro, normotensive rat aorta rings. The data presented demonstrate that aristoteline acts through the opening of potassium channels, inhibition of extracellular calcium influx and activation of the nitric oxide-independent soluble guanylate cyclase. These findings are of limited physiological relevance, since aristoteline causes a significant decrease of vessel tone only at high concentrations. The experiments are logically designed and carefully executed; their presentation, however, can be improved by showing original recordings of, at least, the most important experiments.

Ion channel modulation is claimed on the basis of indirect experimental evidences and, therefore, should be mitigated. Furthermore, additional experiments using more specific pharmacological tools should be performed to finalise the conclusions related to aristoteline-induced effects on cellular Ca2+ homeostasis. I also have some concerns regarding methodology, statistical analysis and interpretation of results. The English style should be carefully revised.

Major Comments

General

1.     The authors assert that “Aristotelia chilensis (Molina) Stuntz 26 (Maqui) has shown important medicinal properties in traditional usage”, that “The fruits and leaves of Aristotelia chilensis (Molina) Stuntz has shown an important range of biological properties including antioxidative, anti-inflammatory and analgesic effects”, etc. These statements should be supported by published literature. New references were included to support the statements in the Introduction section.

Methods

1.                  Several pharmacological tools were used to investigate the targets of aristoteline vasorelaxing activity. As stated in the text, each agent was added to the bath solution several minutes before phenylephrine- or high KCl-induced contraction. First, the effect exerted by each compound on the tone induced by the same concentration of the stimulating agent should be reported. To avoid an increase on the basal tone, the concentrations of the different substances were chosen as low as possible. Each original register was checked, and no significant increases were observed on the basal tone by pre-incubation with L-NAME (10-4 M), potassium blockers (TEA, BaCl2, glibenclamide; 10-5 M), ODQ (10-6 M), indomethacine, or caffeine (10 mM) before the vascular contractile response to KCl or PE. We included more explanations in the Methods section. 

Second, in case the phenylephrine- or KCl-induced tone under the various experimental conditions was different, this might have influenced the subsequent aristoteline-induced vasorelaxation. Keeping this point in mind, the relaxation effect was measured as % the maximal contraction to PE (10-6 M) or KCl (60 mM). While, the vascular contractile response to PE (10-10 to 10-5 M) or KCl (10 to 60 mM) was measured as % of the third curve of the KCl (60 mM).

2.     The use of 4-amino pyridine as a blocker of KV channels might help the analysis of aristoteline mechanism of action. Although we did not used 4-amino pyridine because of the price, we showed that ARI reduced the vascular contractile response to TEA 1 mM. This finding is very interesting because in previous studies, TEA showed an IC50 for Kv1.1 and all members of the Kv3 about 0.1 to 0.3 mM [1].

3.     I am concerned by the use of the same concentration-response curve to aristoteline recorded under control conditions for six different experiments. Was control repeated in parallel with all the six experimental conditions? If this was not the case, results might be severely influenced by a probable inaccuracy due to biological variability. Our equipment consist of a 16 organ baths in total, so it is possible to record simultaneously, in parallel several experimental conditions in duplicate with their respective controls.

4.     Statistical analysis. “Beware of using two-way ANOVA to compare curves or to analyse data where one of the factors is concentration. In fact, the ANOVA calculations ignore the concept of trend” (see Fitting models to biological data. GraphPad Prism). To analyze the trend of the curves, we calculated the pIC50 values using non-lineal regression.

Results

1.     It is clear that aristoteline relaxes aorta ring muscle tone. However, it is necessary to quantify this effect by showing the pIC50 values recorded under the various experimental conditions. Furthermore, introducing the 3 and 30 µM concentrations will aid the correct calculation of this key parameter. We included in the Results the pIC50 values of each curve, and compared the mean of the pIC50 from two different groups with Student’s t-tests.

2.     I suggest to show original recordings of the most significant experiments.  We have included new original recordings.  

3.     Smooth muscle contraction induced by phenylephrine in the absence of extracellular calcium is due to the release of calcium from the intracellular stores through the synthesis of IP3. Restoring extracellular calcium causes an additional contraction due to the influx of calcium through CaV1.2, receptor- and store-operated calcium channels (McFadzean & Gibson, 2002, Br. J. Pharmacol. 135, 1-13). This should be taken into account when hypothesizing the mechanisms underlining the vasorelaxing activity of aristoteline. Thank you for this important reference. We have included this information in the Discussion section.

4.     The best protocol to study the effect of a drug on CaV1.2 channel opening-induced contraction is to construct a concentration-response curve to calcium in rings depolarised with high KCl in the presence of 1 mM EGTA. This protocol resembles that based on the use of Bay K 8644 plus 15 mM KCl. Since the contractile response to KCl (10 to 60 mM) decreased in presence of ARI, but not the sensitivity to KCl (Fig. 2A), this suggests the importance of the influx of calcium from extracellular through CaV1.2 on the vascular contraction. We will consider the use of the high KCl in free calcium medium for further studies.

5.     I am concerned by the use of 10 µM TEA as a concentration which selectively blocks large and low conductivity potassium channels. In fact, TEA is commonly employed at 1 mM concentration to selectively block KCa1.1 channels (previously named BKCa channels) and at 10 mM concentration to unspecifically block potassium channels. Since TEA blocks the KCa1.1 channels with an IC50 of about 200 mM [2,3], a concentration < 1 mM TEA in the relaxation experiment was used to avoid an increase of the basal tone (Fig. 4C). Previous studies showed that 10 mM TEA reduce on quercetin-induced relaxation in the rat aorta [4], or in the presence of glibenclamide the resting potential did not change by 10 mM TEA in pulmorary artery of rabbit [5]. As shown the Fig. 5C, ARI decreased the contractile response to TEA (1 mM), suggesting the participation of the KCa1.1. on the ARI-induced vasodilatation.

We included this explanation in the Discussion.

Discussion

1.     The discussion is very confusing. Some paragraphs lack a proper conclusion (e.g., l. 160-163, l. 179-183), are too speculative (e.g., l. 168-169: only patch-clamp experiments can substantiate this statement) or need to be rephrased (e.g., l. 188-196). In order to gain a clearer insight, the Discussion was rewritten.

Minor Comments:

General

1.     Please, replace “voltage-dependent calcium channels type L” with “CaV1.2 channels” throughout the text. We have corrected as suggested.

Introduction

1.     P. 2 l. 46-48 This sentence is unclear. Please, rephrase. This sentence was rewritten.

2.     P. 2 l. 70-73 These statements are not supported by references. Furthermore, what is “pulse”? These statements have been rewritten to present the information in a clearer way.

 Materials and methods

1. “*P < 0.05, **P < 0.01 …”. You reasonably set P as<0.05. If you want to show <0.01 or 0.001 in the Figures/Tables then you have to set your threshold P as <0.01 and 0.001, respectively, and declare anything outside (i.e., <0.05) as not significant. Use only one level of P, and stick to it. This is a commonplace error. A P-value of<0.05 was considered statistically significant, and we corrected it through the entire manuscript.

2.     Was the Bay K 8644 used a mixture of (+) (calcium antagonist) and (-) (calcium agonist) isomers or the pure (-) isomer? Which solvent was used to dissolve it? Yes, we used (±)-Bay K 8644. A stock solution in DMSO (10−3 M) was prepared with (±)-Bay K 8644 and then, dilutions in KRB were added into the organ bath. We included the full name of (±)-Bay K 8644 in Material and Methods, 4.1. Drugs section.

3.     Subheadings need to be extensively revised as they are not clear. We revised the subheadings in Material and Methods.

4.     P. 8 l. 270-271. Rat aorta rings are not a pharmacological tool. We deleted this statement.

Results

1.     The molecular structure of aristoteline would be helpful to the reader. The molecular structure of aristoteline was included in the Results section.  

2.     P. 2 l. 85. Replace “dose” with “concentration”. We have corrected this word.

3.     Phenylephrine-induced contraction in the absence of extracellular calcium was recorded in the presence of 1 mM EGTA. Was this considered when restoring extracellular calcium? In other words: does the calcium concentration reported in Figure 3B indicate free calcium? The new Fig. 4A (Fig.3A) shows the contractile response to 1 mM PE in a free calcium medium, while the new Fig. 4B (Fig.3B) shows the contractile response after restoring extracellular calcium (0.1 mM to 1 mM). We have written the sentence in the Results to explain the Fig. 4B.

Discussion

1.     P. 6 l. 184. Please, introduce this data in the Result section. We did a new figure with the Indomethacin result on the vasodilation to ARI. And, we also rewrote the statement in the Discussion section.

Conclusion

1.     Please, provide a reference to support the statement on l. 312-313. We have added new references.

Figures

1.     I suggest to indicate the statistical test used in all the legends to Figures.  We have indicated the statistical tests used in all the legends.

2.     Figure 2A. It is not clear why the response to 60 mM KCl is about 130% of the contraction induced by the same concentration of the same agent (as stated on the Y axis). That is because the vascular contractile response to KCl (10 to 60 mM) was measured as % of the third curve of the KCl (60 mM).

Furthermore, it is not clear whether the asterisks refer to caffeine, aristoteline, or both. We have written an explanation in the legend.

3.     Legend to Figure 3. Please indicate the aristoteline incubation time. This information was included in the legend.

4.     Legend to Figure 4. Please, indicate the stimulating agent used to contract vessels. We have completed the text of the legend.

5.     Figure 5. The Y axis titles are not clear. Furthermore, aristoteline incubation time and concentration should be stated in the legend. We have change the Y axis titles for “% Contraction”, and we have a sentence clearer in the legends. And also, the aristoteline incubation time was stated in the legend.

References

1.            Cox, R.H. Molecular determinants of voltage-gated potassium currents in vascular smooth muscle. Cell Biochem Biophys 2005, 42, 167-195, doi:10.1385/CBB:42:2:167.

2.            Nelson, M.T.; Quayle, J.M. Physiological roles and properties of potassium channels in arterial smooth muscle. Am J Physiol 1995, 268, C799-822, doi:10.1152/ajpcell.1995.268.4.C799.

3.            Zhang, H.; Li, P.; Almassi, G.H.; Nicolosi, A.; Olinger, G.N.; Rusch, N.J. Single-channel and functional characteristics of a KCa channel in vascular muscle membranes of human saphenous veins. J Cardiovasc Pharmacol 1996, 28, 611-617.

4.            Satoh, H.; Nishida, S. Cardio-Electopharmacology and Vasodilating Mechanisms of Quercetin. Medicinal Chemistry 2014, 4, 523- 530 doi:10.4172/2161-0444.1000189.

5.            Osipenko, O.N.; Evans, A.M.; Gurney, A.M. Regulation of the resting potential of rabbit pulmonary artery myocytes by a low threshold, O-2-sensing potassium current. British Journal of Pharmacology 1997, 120, 1461-1470, doi:10.1038/sj.bjp.0701075.

Reviewer 2 Report

The manuscript by Fernando Romeroand colleagues deals with the vasorelaxant activity of Aristoteline. It is the continuation of research in the field of biological activity of compounds contained in Aristotelia chilensis. The paper in some way extends our knowledge of the activity of Aristotelia chilensis. The experiments have been well planned and performed, however employing rather simple methods. In my subjective opinion, the paper is interesting but contains a lot of ambiguities that should be explained.

1.     The introduction is not informative enough. It includes some basic information on alkaloids. In my opinion it should contain more precise and current data on the ARI and mechanisms involved in the regulation of vascular tension, which can state the proper background for the discussion. 

2.     The evidence for the purity and solubility of ARI (used in the experiments) should be provided. 

3.     The authors should provide the information regarding the effect of pre-treatment of aortic rings with different drugs ( ODQ, TEA, BaCl2,glibenclamid) on the amplitude of pre-contraction with phenylephrine (1 µM). It is very well known that the inhibition of sGC by ODQ potentiates the responses of contractile agonists in the rat aorta. Considering the fact that the relaxant potency of the vasodilators may change with pre-contraction levels, it is very important to obtain matching contractions before relaxation responses are elicited. Therefore, the authors must confirm that they had elicited matching contractions with phenylephrine in the presence or absence of different treatments before they elicited relaxant responses to ARI (Experimental protocol 4.5.2., 4.5.5.)

4.     The investigators suggest that ARI activated the vascular BKCa and KIR channels in aortas. However, they do not explore the mechanism of the interaction of Ari and potassium channel. The vasorelaxant effect of ARi may be associated with direct or indirect regulation of K+efflux. This hypothesis should be further supported by some experiments definitely conforming the vasorelaxant activity of ARI, such as binding for potassium channels or other available for the authors. There are many suggestions and much less evidence for the mechanism of action of ARI.

5.     The discussion contains many comparisons with some anther alkaloids. In my opinion, the activity of that alkaloids does not support and confirm the activity of ARI. It should be changed. The discussion should be based on the obtained results and should discuss them on the basis of current literature describing the mechanisms of vascular contraction.    

6.     Conclusions should be based on the obtained results rather than describing further research that should be carried out (in the author's opinion). Conclusions do not result from the obtained results and they are not consistent with the paper. 

Author Response

Reviewer 2

The manuscript by Fernando Romero and colleagues deals with the vasorelaxant activity of Aristoteline. It is the continuation of research in the field of biological activity of compounds contained in Aristotelia chilensis. The paper in some way extends our knowledge of the activity of Aristotelia chilensis. The experiments have been well planned and performed, however employing rather simple methods. In my subjective opinion, the paper is interesting but contains a lot of ambiguities that should be explained.

1.     The introduction is not informative enough. It includes some basic information on alkaloids. In my opinion it should contain more precise and current data on the ARI and mechanisms involved in the regulation of vascular tension, which can state the proper background for the discussion. We have written new paragraphs and included new references in the Introduction.

2.     The evidence for the purity and solubility of ARI (used in the experiments) should be provided. We have stated in the Material and Methods, 4.1. Drugs: Aristoteline had a purity higher than 98% (by HPLC with MS detector, NMR and melting point measurements). A stock solution in DMSO (10−3 M) was prepared with ARI and then, dilutions in KRB were added into the organ bath. Final DMSO concentration in the organ bath was lower than 0.1%.

3.     The authors should provide the information regarding the effect of pre-treatment of aortic rings with different drugs (ODQ, TEA, BaCl2, glibenclamide) on the amplitude of pre-contraction with phenylephrine (1 µM). It is very well known that the inhibition of sGC by ODQ potentiates the responses of contractile agonists in the rat aorta. Considering the fact that the relaxant potency of the vasodilators may change with pre-contraction levels, it is very important to obtain matching contractions before relaxation responses are elicited. Therefore, the authors must confirm that they had elicited matching contractions with phenylephrine in the presence or absence of different treatments before they elicited relaxant responses to ARI (Experimental protocol 4.5.2., 4.5.5.). We have explained it further in the Material and Methods, sections 4.6 and 4.9.

“We elicited matching contractions with PE in the presence or absence of L-NAME and ODQ before recording the ARI-induced relaxation.”

“The pre-incubation of aortic rings with low concentrations of different potassium channel blockers, such as TEA (10 mM), BaCl2 (10 mM), glibenclamide (10 mM) did not increase per se the basal vascular tone, nor did the amplitude of pre-contraction with phenylephrine (1 µM).”

4.     The investigators suggest that ARI activated the vascular BKCa and KIR channels in aortas. However, they do not explore the mechanism of the interaction of Ari and potassium channel. The vasorelaxant effect of ARi may be associated with direct or indirect regulation of K+ efflux. This hypothesis should be further supported by some experiments definitely conforming the vasorelaxant activity of ARI, such as binding for potassium channels or other available for the authors. There are many suggestions and much less evidence for the mechanism of action of ARI. We have rewritten the Discussion. 

5.     The discussion contains many comparisons with some anther alkaloids. In my opinion, the activity of that alkaloids does not support and confirm the activity of ARI. It should be changed. The discussion should be based on the obtained results and should discuss them on the basis of current literature describing the mechanisms of vascular contraction. In the Discussion section, we have reduced the non-conforming information, and have added more information that helps describe the comparison with some other alkaloids.

6.     Conclusions should be based on the obtained results rather than describing further research that should be carried out (in the author's opinion). Conclusions do not result from the obtained results and they are not consistent with the paper. We have rewritten the Discussion. 

References

1.            Cox, R.H. Molecular determinants of voltage-gated potassium currents in vascular smooth muscle. Cell Biochem Biophys 2005, 42, 167-195, doi:10.1385/CBB:42:2:167.

2.            Nelson, M.T.; Quayle, J.M. Physiological roles and properties of potassium channels in arterial smooth muscle. Am J Physiol 1995, 268, C799-822, doi:10.1152/ajpcell.1995.268.4.C799.

3.            Zhang, H.; Li, P.; Almassi, G.H.; Nicolosi, A.; Olinger, G.N.; Rusch, N.J. Single-channel and functional characteristics of a KCa channel in vascular muscle membranes of human saphenous veins. J Cardiovasc Pharmacol 1996, 28, 611-617.

4.            Satoh, H.; Nishida, S. Cardio-Electopharmacology and Vasodilating Mechanisms of Quercetin. Medicinal Chemistry 2014, 4, 523- 530 doi:10.4172/2161-0444.1000189.

5.            Osipenko, O.N.; Evans, A.M.; Gurney, A.M. Regulation of the resting potential of rabbit pulmonary artery myocytes by a low threshold, O-2-sensing potassium current. British Journal of Pharmacology 1997, 120, 1461-1470, doi:10.1038/sj.bjp.0701075.

Reviewer 3 Report

General comments:

Romero et al have investigated the vasoreactive effects of aristoteline in isolated rat aorta model.

Major comments:

1) Ex vivo model of isolated aortic rings is indeed a useful system for testing the vasoreactivity potency of different compounds. However, the comprehensive testing of the potential pharmacological potencies in the terms of the future therapeutic applications of various compounds can be achieved only by using the respective in vivo models. Please include this fact in the discussion section, as a clear limitation of ex vivo models.

2) Following the previous line of thinking, is there any plan to perform the experiments with aristoteline using the in vivo models of hypertension? This would be the logical continuation of the study. Please explain and if this plan is the truth, I would expect some words about it in the discussion section.

3) Interestingly, this study comes from Chile, a country with significant mountain regions and many people working in hypoxic conditions. Therefore, it would be interesting to investigate the vasodilator properties of aristoteline on hypoxic pulmonary vasoconstriction. If possible, I would strongly suggest additional experiments with isolated pulmonary arteries exposed to hypoxic condition?

4) Finally, what was the ultimate aim of your study? Did you analyze the vasoreactive effects having in mind the future use of this indole-alkaloid against hypertensive disorders or not? Please include some text parts in the discussion with focus on potential future use of this substance.

Minor comments:

1) Please check again the whole text for the typos. For example, in the Introduction section, there are 2 times written word “including”: “…of biological properties including including antioxidant…”

2) Similarly, another typo appears in the Conclusions section: “…natural alkaloid aristoteline obtained from of Aristotelia…” Please correct.

Round  2

Reviewer 1 Report

Molecules, Manuscript ID: molecules-443696 Revised Version

This manuscript has been extensively, though not satisfactorily revised.

Several points raised in the previous review were not answered or only partly addressed and the additional experiments required were not performed.

Statistical analysis (i.e. non-linear regression) giving rise to pIC50 or pEC50 values has been performed but the data included in the text are meaningless.

Finally, the language quality is sometimes so poor that it is difficult to understand.

Major Comments

Methods

1.     … Second, in case the phenylephrine- or KCl-induced tone under the various experimental conditions was different, this might have influenced the subsequent aristoteline-induced vasorelaxation.

R. Keeping this point in mind, the relaxation effect was measured as % the maximal contraction to PE (10-6 M) or KCl (60 mM).

This is not correct, as relaxation is markedly influenced by the level of tension developed by the tissue, and simple normalization does not correct for this error.

2.     The use of 4-amino pyridine as a blocker of KV channels might help the analysis of aristoteline mechanism of action.

R. Although we did not used 4-amino pyridine because of the price, we showed that ARI reduced the vascular contractile response to TEA 1 mM. This finding is very interesting because in previous studies, TEA showed an IC50 for Kv1.1 and all members of the Kv3 about 0.1 to 0.3 mM [1].

Given that 25 g of 4-AP cost around 40 €, according to the reference cited ([1]), 4-AP remains the gold standard for studying KV channels; moreover, KV3 channels seem not to be present in rat aorta (see Table 5 in the same reference).

Results

1.     The pIC50 or pEC50 values reported in the revised version of the manuscript are only mean values with no errors and neither have any meaning in terms of pIC50 nor correspond to the matching concentration-response curve. For example, in Figure 2C pIC50 values of 1.30 µM and 1.56 µM were given for control and ODQ curves, respectively. These values are neither the -log of the IC50 value nor IC50 values. In fact, the ODQ curve shows that 10 µM ARI caused about 10% relaxation, while the control curve, at the same concentration, about 40% relaxation. Thus, it follows that IC50 values must be greater than 10 µM. Moreover, how were mean values compared by Student’s t test if errors, such as s.e.m., have not been calculated? Additionally, pIC50 was confused with pEC50 (e.g. Figure 4). Finally, the suggestion to introduce the 3 and 30 µM concentrations to aid the correct calculation of these key parameters was ignored by the authors.

2.     As stated by the authors, “TEA blocks the KCa1.1 channels with an IC50 of about 200 mM (Nelson & Quayle, 1995)”. First of all, the value is 200 µM. Therefore, 10 µM TEA is well below this key value and thus inadequate to reach any kind of conclusion from the experiment shown in the new Figure 6C.

Minor Comments:

Introduction

1.     P. 2 l. 52 What is “NO injury”?

2.     P. 2 l. 72. Kaempferol, galangin, chrysin and genistein are not alkaloids but rather flavonoids.

Materials and methods

1.     Statistical analysis of Figure 8. Paired Student’s t test can be applied only when control and “aristoteline” experiments are performed on the same ring. According to Figure 7, they have been performed on two different preparations. Therefore, unpaired Student’s t test is more appropriate.

2.     If you prepared a stock solution 1 mM of aristoteline in DMSO (P. 8 l. 248), when you added 10 µM and 100 µM concentrations into the organ bath, solvent concentration was 1% and 10%, respectively, rather than below 0.1%, as stated on l. 249-250.

3.     P. 10 l. 311. “protective”? Furthermore, KCl is missing.

Results

1.     P. 4 l. 131. What does “The efficacy of the maximum tonic response” mean?

2.     P. 5 l. 151-153. The statistical difference claimed by the authors is neither accompanied by asterisks in Figure 6B nor supported by a control pIC50 value.

3.     Figure 7. The marked decrease in passive tone following washout of 60 mM KCl is quite unusual. The authors should give an explanation for this.

Discussion

1.     P. 7 l. 217. It is not clear why aristoteline did not reduce the “sensitivity” to KCl-induced plasma membrane depolarization. Figure 4A shows a significant reduction of this type of contraction.

2.     P. 8 l. 226. Caffeine does not inhibit IP3 receptors but activates ryanodine receptors and inhibits phosphodiesterases.

Figures

1.     P. 3 l. 101 and throughout the text. “4PL” is an acronym not commonly used in the literature.

2.     Figure 4B, 6A-C and 9. The X axis should be entitled “Log [drug] (M)”.

3.     Figure 4B. Why the curve of caffeine is without asterisks?

4.     Legend to Figure 5. Panel 5A shows the effect of aristoteline on Ca2+ release from the phenylephrine-sensitive, intracellular Ca2+ stores and not “on the calcium current blockage in calcium-free medium”.

References

1.     I am not sure whether reference [1] is pertinent to this manuscript.

2.     Reference 5 and 6 are the same.

3.     Reference 19: it is not clear how this paper may be linked to the sentence of P. 2 l. 66-68.

Reviewer 2 Report

The manuscript has been properly changed and improved. The current version is worth publishing in the journal.